# A Method of Setting the LiDAR Field of View in NDT Relocation Based on ROI

**DOI:** 10.3390/s23020843

**Published:** 2023-01-11

**Authors:** Jian Gu, Yubin Lan, Fanxia Kong, Lei Liu, Haozheng Sun, Jie Liu, Lili Yi

**Affiliations:** 1School of Agricultural Engineering and Food Science, Shandong University of Technology, Zibo 255000, China; 2Shandong University of Technology Sub-Center of National Center for International Collaboration Research on Precision Agricultural Aviation Pesticide Spraying Technology, National Sub-Center for International Collaboration Research on Precision Agricultural Aviation Pesticide Spraying Technology, Zibo 255000, China; 3National Center for International Collaboration Research on Precision Agricultural Aviation Pesticides Spraying Technology, College of Engineering, Shandong University of Technology, Guangzhou 510642, China

**Keywords:** normal distribution transformation, high-precision maps, automatic navigation

## Abstract

LiDAR placement and field of view selection play a role in detecting the relative position and pose of vehicles in relocation maps based on high-precision map automatic navigation. When the LiDAR field of view is obscured or the LiDAR position is misplaced, this can easily lead to loss of repositioning or low repositioning accuracy. In this paper, a method of LiDAR layout and field of view selection based on high-precision map normal distribution transformation (NDT) relocation is proposed to solve the problem of large NDT relocation error and position loss when the occlusion field of view is too large. To simulate the real placement environment and the LiDAR obstructed by obstacles, the ROI algorithm is used to cut LiDAR point clouds and to obtain LiDAR point cloud data of different sizes. The cut point cloud data is first downsampled and then relocated. The downsampling points for NDT relocation are recorded as valid matching points. The direction and angle settings of the LiDAR point cloud data are optimized using RMSE values and valid matching points. The results show that in the urban scene with complex road conditions, there are more front and rear matching points than left and right matching points within the unit angle. The more matching points of the NDT relocation algorithm there are, the higher the relocation accuracy. Increasing the front and rear LiDAR field of view prevents the loss of repositioning. The relocation accuracy can be improved by increasing the left and right LiDAR field of view.

## 1. Introduction

Over the past two decades, relocation has played a crucial role in automotive driving applications. Relocation is used to obtain the vehicle’s global position and pose based on the built map. Various relocation methods have been successful in resolving such problems using global navigation satellite systems (GNSS) [1,2,3], the inertial measurement unit (IMU) [4,5], cameras, LiDAR, and other sensing sensors.

The integration of GNSS and IMU is a common solution to these problems [6,7,8]. However, GNSS is prone to signal interference or loss, in urban buildings, for example. The current solution is to use mileage information to compensate for GNSS measurements. However, performance in auto-driving related applications is still generally insufficient.

Cameras are one of the most attractive relocation sensors because of their inherent high information content, low cost, and small size [9,10,11]. Visual relocation uses the large amount of information provided by the camera to estimate the robot position. Because accurate and reliable positioning is required where the GPS signal is weak, a new visual measurement framework for land vehicle positioning was proposed by Zhiyong Zheng [12]. However, for outdoor large-scale relocation, ensuring its robust operation is still very challenging especially in changing environments and adverse weather conditions. Kaixin Yang presents a coordinated positioning strategy that is composed of semantic information and probabilistic data association, which improves the accuracy of SLAM (Simultaneous Localization and Mapping) in dynamic traffic settings [13]. However, many walls and windows in a city are made of glass or vinyl and are easily exposed to light during the day.

An increasingly popular solution to the city problem is to stop using GNSS and camera measurements altogether and rely on LiDAR, which measures 3D scans of the environment. Compared with cameras and GNSS, LiDAR has better penetrability and anti-interference characteristics. LiDAR detects objects and surfaces using distance measurements, which are computed from the time-of-flight of the reflected light pulses [14,15,16]. Furthermore, the amount of data acquired by a LiDAR system is less than in the case of a camera. Hence the LiDAR sensor is used frequently for positioning and object detection. However, most LiDAR-based localization solutions with prior point-cloud maps assume that the road scenes are relatively constant, therefore new constructions, road-side vegetation, partial occlusions by changing objects may severely compromise its robustness. Therefore, an interesting but open question is whether LiDAR can be used for robust relocalization in large-scale changing environments.

In large-scale and dynamic urban environments, the normal distribution transformation (NDT) algorithm based on high-precision maps is one of the main repositioning algorithms widely used in automotive driving applications [17,18]. In comparison to the SLAM algorithm, the NDT algorithm is found to be more effective for large point clouds. The NDT registration algorithm is time-consuming and stable, has little correlation with the initial value, and can be corrected well when the initial error is large. The open source Autoware algorithm at Nagoya University mainly uses the NDT algorithm. The Autoware algorithm uses the NDT matching registration algorithm to obtain its own position and position information in the LiDAR point cloud map. However, the NDT algorithm is not robust regarding significant geometric changes to the environment or overly unexpected or dynamic objects [19,20]. These shortcomings seriously affect the performance of scan matching based on high-precision maps. The NDT relocation algorithm can handle some environmental changes, such as the accidental obscuring of objects (LiDAR sensors partially obscured by leaves or other vehicles), which is unavoidable in cities. When there is a difference between the environment and the map, the positioning accuracy of nondestructive testing decreases. In practical applications multi-LiDAR is used for positioning. The position of the LiDAR directly determines the LiDAR field of view. Different views of the LiDAR field of view have a direct impact on the positioning accuracy of vehicles. Many studies have not specifically assessed the relationship between point cloud occlusion at different locations and noninvasive detection performance based on high-precision maps.

## 2. Algorithm Principle

### 2.1. Loam

LOAM (lidar odometry and mapping in real time) is a high-precision and real-time positioning and mapping algorithm based on 3D LiDAR proposed by Ji Zhang et al. [21,22]. The core of the LOAM framework is in two parts, a high frequency odometer and low frequency mapping, as shown in Figure 1.

The odometer performs scan-scan matching through a high frequency and low number of point clouds, estimates the motion relationship between two frames, and outputs the results to the mapping algorithm; mapping matches and aligns the undistorted point cloud to the map at a frequency of 1 Hz, using the scan-map matching method [23,24]. Finally, the attitude transformation created by the two algorithms is integrated to obtain the transformation output of LiDAR, with an attitude to the map of about 10 Hz.

We start with the extraction of feature points from the LiDAR cloud, $p_{k}$. We select feature points that are on sharp edges and planar surface patches. Let *i* be a point in $p_{k}$, $i\in p_{k}$ and let *S* be the set of consecutive points of *i* returned by the LiDAR scanner in the same scan. This defines a term to evaluate the smoothness of the local surface as
(1)$$c=\frac{1}{\left|S\right|\;\;\;\cdot \;\;\;∥X_{\left(K,i\right)}^{L}∥}∥\sum_{j\in S,j\neq i}\left(X_{\left(K,i\right)}^{L}-X_{\left(K,j\right)}^{L}\right)∥.$$

In this paper, the curvature of a point is calculated according to the above formula. In practice, we only need to compare the curvature of a point so that we can find the curvature of the point in the square coordinate of the difference of five points around a point. In this way, we can find the curvature, *c*, of each point, and by comparing the curvatures, we can select the edge points with a larger curvature and the plane points with a smaller curvature. To prevent the feature points from clustering, each scanned point cloud is divided into four parts, from which two points with the largest curvature are selected as edge points and four points with the smallest curvature are selected as plane points.

When selecting points, we want to avoid selecting points around already selected points or points whose LiDAR lines are close to parallel planes, which are generally considered unreliable because they cannot be seen at any time. We also want to avoid possible obscuring points. The odometry algorithm estimates the motion of the LiDAR within a sweep. Let $t_{k}$ be the starting time of a sweep, *k*. At the end of each sweep, the point cloud perceived during the sweep, $p_{k}$, is reprojected to the time stamp $t_{k+1}$. We denote the reprojected point cloud as ${\bar{p}}_{k}$. During the next sweep, k+1, ${\bar{p}}_{k}$ is used together with the newly received point cloud, $p_{k+1}$, to estimate the motion of the LiDAR. The next step is to find the corresponding relationship, i.e., to match the feature points of two point clouds; the corner point of $p_{k+1}$ matches the corner line of ${\bar{p}}_{k}$, and the plane point of $p_{k+1}$ matches the plane of ${\bar{p}}_{k}$. With the corresponding relationship between point-to-line and point-to-face, we can calculate the distance between point-to-line and point-to-face:(2)$$d_{E}=\frac{\left|\left({\tilde{X}}_{\left(k+1,i\right)}^{L}-{\bar{X}}_{\left(k,j\right)}^{L}\right)\times \left({\tilde{X}}_{\left(k+1,i\right)}^{L}-{\bar{X}}_{\left(k,l\right)}^{L}\right)\right|}{\left|\left({\bar{X}}_{\left(k,j\right)}^{L}-{\bar{X}}_{\left(k,l\right)}^{L}\right)\right|}$$

Then we can find the distance from the plane point to the corresponding plane:(3)$$d_{H}\frac{\left|\begin{array}{c} \left({\tilde{X}}_{\left(k+1,i\right)}^{L}-{\bar{X}}_{\left(k,j\right)}^{L}\right)\\ \left(\left({\bar{X}}_{\left(k,j\right)}^{L}-{\bar{X}}_{\left(k,l\right)}^{L}\right)\times \left({\bar{X}}_{\left(k,j\right)}^{L}-{\bar{X}}_{\left(k,m\right)}^{L}\right)\right) \end{array}\right|}{\left|\left({\bar{X}}_{\left(k,j\right)}^{L}-{\bar{X}}_{\left(k,l\right)}^{L}\right)\times \left({\bar{X}}_{\left(k,j\right)}^{L}-{\bar{X}}_{\left(k,m\right)}^{L}\right)\right|}$$

The LiDAR motion is modelled with constant angular and linear velocities during a sweep. This allows us to linearly interpolate the pose transformation within a sweep for the points that are received at different times. If we let *t* be the current time stamp, and recall that $t_{k+1}$ is the starting time of sweep k+1, the linear interpolation formula is
(4)$$T_{\left(k+1,i\right)}^{L}=\frac{t_{i}-t_{k+1}}{t-t_{k+1}}T_{k+1}^{L}.$$

In order to obtain the corresponding relationship between the points in this frame data and the points in the previous frame data, we use a rotation matrix *R* and a translation amount *T*.
(5)$$X_{\left(k+1,i\right)}^{L}=R{\tilde{X}}_{\left(k+1,i\right)}^{L}+T_{\left(k+1,i\right)}^{L}\left(1:3\right)$$

Since the derivation of rotation matrix is very complex, the rotation matrix *R* is expanded as follows by the Rodrigues formula:(6)$$R=e^{\hat{\omega }\theta }=I+\hat{\omega }\sin\theta +{\hat{\omega }}^{2}\left(1-\cos\theta \right).$$

This makes it easy to derive the rotation matrix.

Now we have the distance from point-to-line and point-to-face, we can obtain the error function for optimization:(7)$$f\left(T_{k+1}^{L}\right)=d.$$

Each line in *f* represents a characteristic point. The next requirement is to solve the Jacobian matrix. Finally, the LM method is used for optimization:(8)$$T_{k+1}^{L}\leftarrow T_{k+1}^{L}-{\left(J^{T}J+\lambda diag\left(J^{T}J\right)\right)}^{-1}J^{T}d$$

Since the solution in the previous step is the result according to the local LiDAR observation coordinate system $T^{L}$, it solves the transformation between adjacent frames. However, in order to simultaneously locate and map, it is necessary to solve the transformation under the global coordinate system $T^{W}$. Therefore, when we obtain the attitude transformation information of several adjacent frames, we need to match it with the global map and add it to the global map.

Finally, the attitude information obtained from the LiDAR odometer solution and the information obtained from the map construction are transformed and integrated, through the use of rviz software, for example.

### 2.2. NDT Relocation Algorithm

In order to identify the location of the LiDAR in the offline map, we compared the point cloud from the LiDAR scan with the point cloud from the offline map. During the relocation process, the point cloud from the LiDAR scan may differ from the point cloud from the offline map, either because the LiDAR field of view is occluded, or because the vehicle uses only part of the LiDAR point cloud.

For relocation in maps with deviations, we use the NDT alignment algorithm, which does not compare the difference between two point clouds but transforms the reference point cloud map into a normal distribution of multidimensional variables [25,26,27]. If the transformation parameters enable a good match between the two sets of LiDAR data, then the probability density of the transformed points in the reference system will be large. Therefore, an optimization method can be considered to find the transformation parameter that maximizes the sum of the probability densities, when the two sets of LiDAR point cloud data will match best.

The first step is to grid the 3D offline point cloud map, using a small cube to divide the entire space of scanned points, and for each grid, calculate its probability density function based on the points within the grid. This can be described as:(9)$$\vec{\mu }=\frac{1}{m}\sum_{q=1}^{m}{\vec{y}}_{q}$$
(10)$$\Sigma =\frac{1}{m}\sum_{q=1}^{m}\left({\vec{y}}_{q}-\vec{\mu }\right){\left({\vec{y}}_{q}-\vec{\mu }\right)}^{T}$$
where $\vec{\mu }\;$ is the mean of the normal distribution of the grids of the offline map, *m* indicates the number of points in the offline map grid, *q* means the *q*th point in the offline map grid, ${\vec{y}\;}_{q=1,...,m}$ for all scanned points in the offline map grid, and ∑ denotes the covariance matrix of the offline map grid. The probability density function of a grid can be described as:(11)$$f\left(\vec{x}\right)=\frac{1}{{\left(2\pi \right)}^{\frac{3}{2}}\sqrt{\left|\Sigma \right|}}e^{-\frac{{\left(\vec{x}-\vec{\mu }\right)}^{T}\Sigma {}^{-1}\left(\vec{x}-\vec{\mu }\right)}{2}}.$$

The use of normal distribution to represent an otherwise discrete offline point cloud map has many benefits. This chunked, smooth representation is continuously derivable and the probability density function of each lattice can be thought of as an approximation to a local surface, which not only describes the location of the surface in space, but also contains information about the orientation and smoothness of the surface.

When using NDT alignment, the goal is to find the pose of the current LiDAR scan in such a way as to maximize the likelihood of the currently scanned points lying on the surface of the offline map. The parameter we then need to optimize is the transformation (rotation, translation, etc.) of the currently scanned LiDAR point cloud, which we describe using a transformation parameter $\vec{h}\;$. The current scan is a point cloud $X=\{{\vec{x}}_{1},\ldots ,{\vec{x}}_{n}\}$, given the set of scan points *X* and the transformation parameter $\vec{h}\;$, such that the spatial transformation function $T(\vec{h}\;,{\vec{x}}_{q})$ denotes the use of the pose transformation $\vec{h}\;$ to move the points ${\vec{x}}_{q}$, combined with the previous set of density-of-state functions (Probability Density Function for each grid), then the best transformation parameter $\vec{h}\;$ should be the pose transformation that maximizes the likelihood function:(12)$$Likelihood:\Theta =\prod_{q=1}^{n}f\left(T\left(\vec{h},{\vec{x}}_{q}\right)\right).$$

Then, maximizing the likelihood is also equivalent to minimizing the negative log-likelihood −logΘ;
(13)$$-\log\Theta =-\sum_{q=1}^{n}\log\left(f\left(T\left(\vec{h},{\vec{x}}_{q}\right)\right)\right).$$

An optimization algorithm is then used to tune the transformation parameter $\vec{h}\;$ to minimize this negative log likelihood. YThe NDT algorithm uses Newton’s method for parameter optimization. Here the probability density function $f(\vec{x}\;)$ does not have to be normally distributed; any probability density function that reflects the structural information of the scanned surface and is robust to anomalous scan points will be sufficient.

### 2.3. Point Cloud Data Preprocessing

The positive direction of the x-axis is the front of the LiDAR, and the positive direction of the y-axis is the left side of the LiDAR, as shown in Figure 2. Different LiDAR point cloud areas are extracted through the ROI (area of interest), and LiDAR point cloud areas of different sizes are reserved at the front, back, left, and right to simulate the changes of the LiDAR field of view in different degrees. The ROI can be delineated for further processing. The LiDAR point clouds in each region can be described as:
(14)$$f_{f}\left(\alpha \right)=\left\{\begin{array}{c} \left[\left(a,b\right)\left|b-a\tan\beta <0\right.\right]⋂\left[\left(a,b\right)\left|b+a\tan\beta >0\right.\right]\begin{array}{cc} , & \alpha \in \left[0^{\circ },180^{\circ }\right] \end{array}\\ \left[\left(a,b\right)\left|b-a\tan\beta <0\right.\right]⋃\left[\left(a,b\right)\left|b+a\tan\beta >0\right.\right]\begin{array}{cc} , & \alpha \in \left(\left.180^{\circ },360^{\circ }\right]\right. \end{array} \end{array}\right.,$$
(15)$$f_{b}\left(\alpha \right)=\left\{\begin{array}{c} \left[\left(a,b\right)\left|b-a\tan\beta >0\right.\right]⋂\left[\left(a,b\right)\left|b+a\tan\beta <0\right.\right]\begin{array}{cc} , & \alpha \in \left[0^{\circ },180^{\circ }\right] \end{array}\\ \left[\left(a,b\right)\left|b-a\tan\beta >0\right.\right]⋃\left[\left(a,b\right)\left|b+a\tan\beta <0\right.\right]\begin{array}{cc} , & \alpha \in \left(\left.180^{\circ },360^{\circ }\right]\right. \end{array} \end{array}\right.,$$
(16)$$f_{l}\left(\alpha \right)=\left\{\begin{array}{c} \left[\left(a,b\right)\left|b-a\tan\beta >0\right.\right]⋂\left[\left(a,b\right)\left|b+a\tan\beta >0\right.\right]\begin{array}{cc} , & \alpha \in \left[0^{\circ },180^{\circ }\right] \end{array}\\ \left[\left(a,b\right)\left|b-a\tan\beta >0\right.\right]⋃\left[\left(a,b\right)\left|b+a\tan\beta >0\right.\right]\begin{array}{cc} , & \alpha \in \left(\left.180^{\circ },360^{\circ }\right]\right. \end{array} \end{array}\right.,$$
(17)$$f_{r}\left(\alpha \right)=\left\{\begin{array}{c} \left[\left(a,b\right)\left|b-a\tan\beta <0\right.\right]⋂\left[\left(a,b\right)\left|b+a\tan\beta <0\right.\right]\begin{array}{cc} , & \alpha \in \left[0^{\circ },180^{\circ }\right] \end{array}\\ \left[\left(a,b\right)\left|b-a\tan\beta <0\right.\right]⋃\left[\left(a,b\right)\left|b+a\tan\beta <0\right.\right]\begin{array}{cc} , & \alpha \in \left(\left.180^{\circ },360^{\circ }\right]\right. \end{array} \end{array}\right..$$

The front, back, left, and right of the LiDAR point cloud areas are represented by $f_{f}\left(\alpha \right)$, $f_{b}\left(\alpha \right)$, $f_{l}\left(\alpha \right)$, $f_{r}\left(\alpha \right)$. The point inside the LiDAR point cloud is represented by (a,b), α is the angle of view of the cut LiDAR point cloud, and β is from $0^{\circ }$ to $90^{\circ }$, as shown in Figure 3.

Voxel downsampling creates a 3D voxel grid (considering the voxel grid as a collection of spatial 3D cubes) from the input point cloud data [28,29,30,31]. Then within each voxel (3D cube), the other points in the voxel are approximated by the center of gravity of all points in the voxel, so that all points in the voxel are represented by one center-of-gravity point. Voxel downsampling creates a 3D voxel grid (considering the voxel grid as a collection of spatial 3D cubes) from the input point cloud data. Then within each voxel (3D cube), all points in the voxel are approximated using the voxel’s center of gravity to reveal other points in the voxel so that all points in the voxel are represented by a single center of gravity point. As a result, the number of points is reduced, the matching speed is improved, the shape features of the point cloud remain basically unchanged, and the spatial structure information is preserved. The larger the voxel grid selection, the smaller the sampled point cloud, and the faster the processing speed; however, the original point cloud will be too blurred. A smaller voxel grid selection will have the opposite effect. At the same time, it is necessary to record the number of points of the different LiDAR point cloud data after desampling, as shown in Figure 4.

## 3. The KITTI Dataset Test

To test the effect of the different fields of view of LiDAR on the NDT relocalization algorithm, we used the KITTI dataset with a full length of 864.831 m and a duration of 117 s. The test platform was a Velodyne HDL-64E-equipped vehicle. All experiments were performed on this platform. The average speed of the vehicle was about 2.5 m/s. All the evaluation experiments were run on a computer with an AMD R7-4800H processor, 16 GB RAM, and a single NVIDIA GeForce GTX1650ti GPU. Velodyne HDL-64E is a 64-line digital LiDAR mounted directly above the mobile chassis, with a 360° horizontal field of view, 5–15 Hz rotational speed, a 26.8° vertical field of view (+2° to −24.8°), vertical angular resolution of 0.4°, horizontal angular resolution of 0.08°, a point cloud count up to 1.3 million points per second, a maximum range of 100 m, and a ranging accuracy of ±2 cm.

As shown in Figure 5, a high-precision map is constructed by using the loam algorithm for the KITTI dataset. The NDT matching points are the points where the original point cloud data has been ROI processed and downsampled, as shown in Figure 6. Due to the uneven density of the original point cloud after voxel downsampling, the number of NDT matching points in each direction is also irregular, so the positioning accuracy of the NDT matching points in each direction is different. The downsampling factor set by the code in this article is 3.0. The transformation epsilon is 0.05, the step size is 0.1, the resolution is 2.0, and the maximum number of iterations is 30. This paper uses root mean squared error (RMSE) to measure the positioning accuracy. The RMSE is the square of the ratio of the square of the deviation between the predicted value and the true value and the number of observations.

As shown in Table 1, the RMSE obtained from the LiDAR point cloud reposition tracks at 90°, 180°, and 270° in different directions is compared. It can be seen that in the same direction, the larger the view angle of the LiDAR and the more effective the matching points of the NDT algorithm, the higher the positioning accuracy and the smaller the error. The LiDAR cannot be repositioned when the left and right LiDAR field of view angles are 90 degrees. When the front and back LiDAR field of view angle is 90°, it can be repositioned. When the angle of the LiDAR field of view in four directions is enlarged, the precision of the front and back sides is improved greatly, while the precision of the left and right sides is improved less.

In order to further explore the effect of the LiDAR-matching point cloud orientation and the number of NDT matching points on the positioning accuracy and weight in the point cloud, the influence of the positioning trajectory drift is extended in the middle of 90° to 270° on the left and right. Experiments are carried out at different angles at 30 degrees. As shown in Table 2, by comparing the number of NDT matching points on the left and right sides of the LiDAR, it can be seen that due to the large number of NDT-matching point clouds at the front and the back, and at 90°–270° on the left and right sides, at 1°, the proportion of the number of matching points before and after NDT plays a decisive role in the accuracy of the positioning trajectory. At the same time, it is also proved that the more NDT matching points there are, the higher the positioning accuracy and the smaller the degree of drift. In normal driving mode, since the point clouds at the front and back of the radar point cloud are relatively abundant, it is recommended to use the front and back point clouds more.

Based on the squeezing theorem and a large number of experiments, and setting the LiDAR angle accuracy to 1°, the critical value of the relocation loss in four directions is obtained. When NDT matching points are 21.3% to 24.9% of the total, there is a high probability of missing NDT relocations. The limit is between 137,933 and 161,495. The results are shown in Table 3.

With an approximate number of NDT matching points, only a smaller LiDAR field of view angle is required to use the previous NDT matching points. Front NDT matching points also have larger errors. This reflects the robustness of the NDT matching points at the front and back ends.

## 4. Urban Dataset Testing

Tracked vehicles equipped with 64-line LiDAR are used in real environments, as shown in Figure 7. The LiDAR used here was the Ouster OS1-64, with a measurement range of 150 m, an accuracy of ±2 cm, a vertical perspective of 30°, a horizontal perspective of 360°, a vertical angle resolution of 0.52°, a horizontal angle resolution of 0.09°, and a rotation rate of 10 Hz. The test was conducted in a complex urban environment with obvious characteristics. As shown in Figure 8, there were curved roads, straight roads, and obvious large obstacles on the left and right sides of the crawler. The road length in the dataset was 1 s and the duration was 23 s. All evaluation experiments were run on a computer with an AMD R7-4800H processor, 16 GB RAM, and a single NVIDIA GeForce GTX1650ti GPU.

It can be seen from Figure 9 and Figure 10 that when the number of NDT matching points is less than 100, the relocalization error will increase sharply, and when the number of NDT matching points is less than 50, it is extremely easy to lose positioning. The theoretical value for the number of NDT matches per frame is set to 100. Since the number of NDT matching points is 3109, the number of NDT matching points is 249,2189. Therefore, the number of NDT valid matches cannot be less than 12.4% of the total number, and the number of single-frame NDT valid matches cannot be less than 100. However, the complexity of the road sections (there are straight sections and curved sections) led to the fluctuation of NDT matching points in terms of the volume of the point cloud frame. According to the actual experiment, a reliable range was obtained, which was 14.7% to 16.3%, as shown in Table 4.

It can be seen from Figure 9 and Figure 10 that each group of data has four turning points with great changes. As shown in Figure 11 and Figure 12, as the vehicle turns, the number of LiDAR-effect matching points on both sides decreases dramatically, resulting in reduced repositioning accuracy, as shown in Figure 13 and Figure 14. The LiDAR field of view on the left and right sides becomes larger, and the number of effective matching points increases dramatically, resulting in higher repositioning accuracy, as shown in Figure 15 and Figure 16.

As shown in Figure 17 and Figure 18, in the normal, straight-line segment, obstacles were distributed unevenly and close by. It is easy to reduce the left and right LiDAR field of vision, resulting in a number of NDT-effective matching points between 50 and 100, resulting in reduced relocation accuracy, or even loss of location. In the normal, straight-line segment, since most of the front and back LiDAR data were ground points, the relocation accuracy was seriously affected. The left and right LiDAR points were mostly effective feature points with rich features. As shown in Figure 19 and Figure 20, in the straight-line segment, the positioning accuracy of the LiDAR data on the front and back sides was poor. As shown in Figure 21 and Figure 22, the positioning effect of the LiDAR data on the left and right sides was good.

## 5. Conclusions

This paper presents a LiDAR layout and field-of-view selection method based on high-precision map NDT relocation to solve the problem of large NDT relocation error and position loss when the field of view is too large to be obstructed. In order to simulate the real placement environment and obstructed LiDAR, an ROI algorithm is used to cut the LiDAR point cloud to obtain different sized LiDAR point cloud data. First, the cut point cloud data is downsampled, then relocated. The downsampling points for NDT relocation are recorded as valid matching points. The direction and angle settings for LiDAR point cloud data are optimized using RMSE values and valid matching points. The results show that in urban scenes with complex road conditions, there are more front and back matching points than left and right matching points in the unit angle. The more matching points the NDT relocation algorithm has, the higher the relocation accuracy will be. Increasing the front and back LiDAR field of view prevents the loss of repositioning. The effective matching points of single-frame LiDAR data are larger than the set threshold. By increasing the field of view of the left and right LiDAR, the repositioning accuracy can be improved. You also need to keep a safe distance from obstacles on both sides. The future research plan is to improve the NDT algorithm to speed up its processing speed, increase the positioning accuracy, and enable it to be relocated, even when the LiDAR data is sparse.

## Figures and Tables

**Figure 1 sensors-23-00843-f001:**
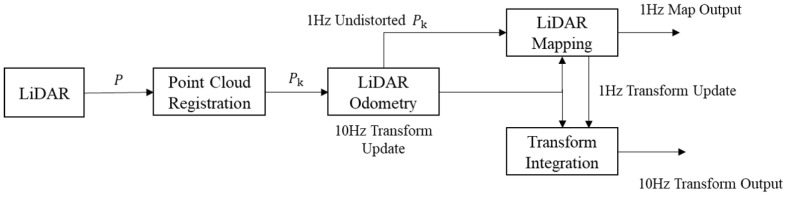
System structure diagram of the LOAM algorithm.

**Figure 2 sensors-23-00843-f002:**
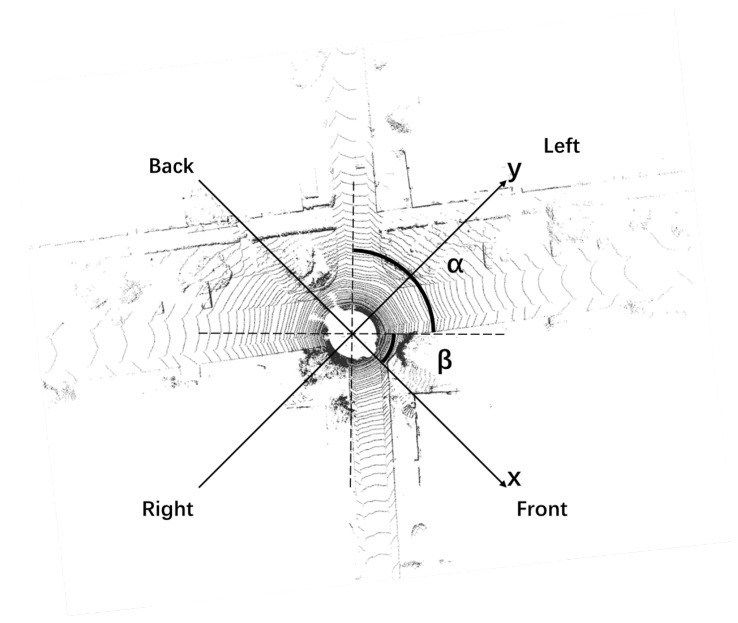
Simulation of different fields of view of LiDAR.

**Figure 3 sensors-23-00843-f003:**
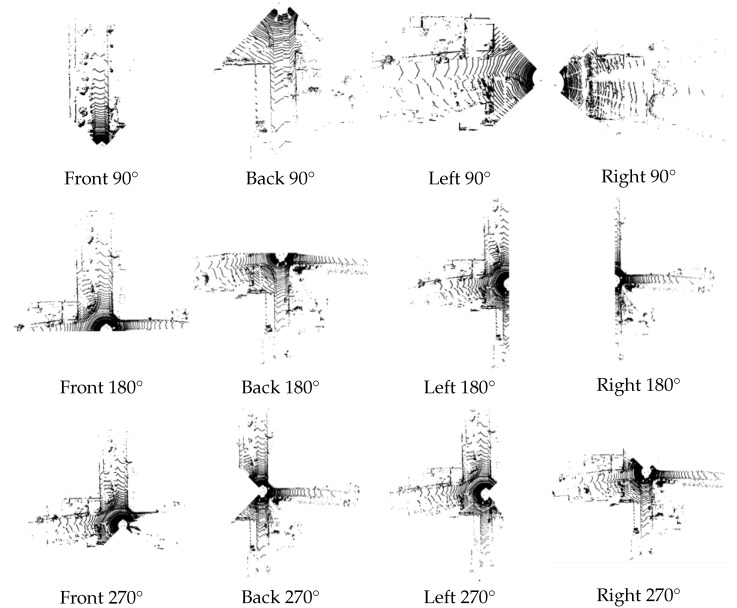
Simulation of the different fields of view of LiDAR.

**Figure 4 sensors-23-00843-f004:**
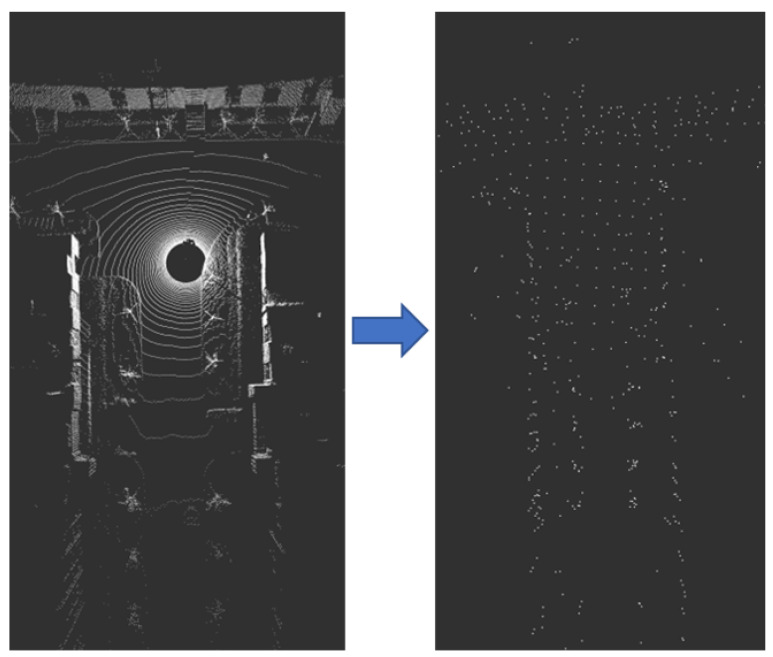
LiDAR point cloud data downsampling.

**Figure 5 sensors-23-00843-f005:**
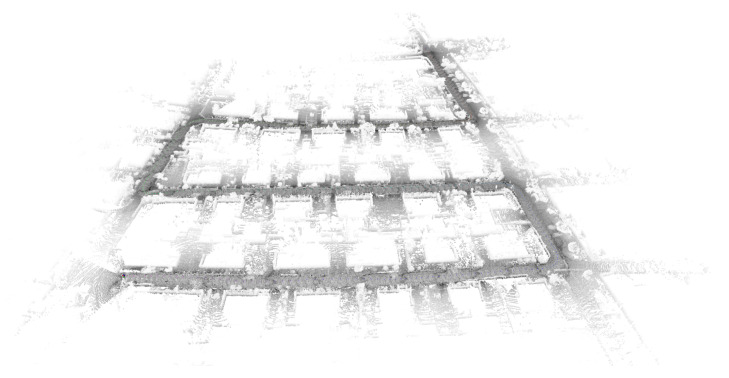
The high-precision maps of KITTI datasets.

**Figure 6 sensors-23-00843-f006:**
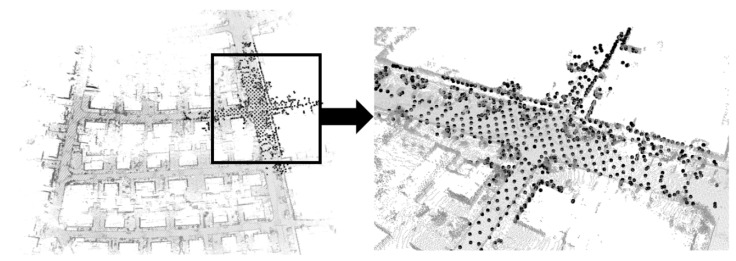
Local magnification during NDT relocation in KITTI datasets.

**Figure 7 sensors-23-00843-f007:**
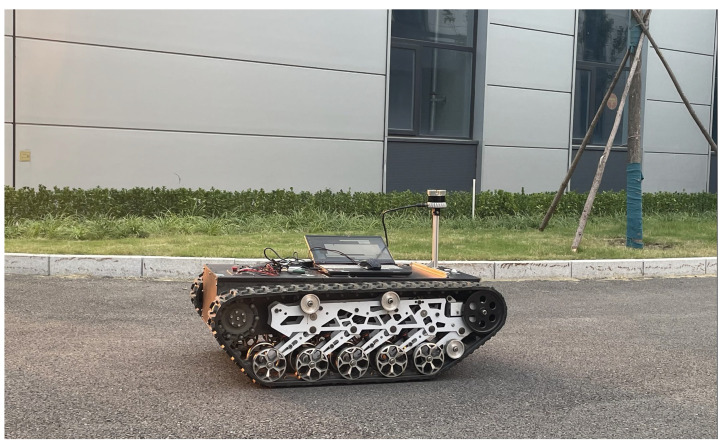
Tracked vehicles equipped with 64-line LiDAR.

**Figure 8 sensors-23-00843-f008:**
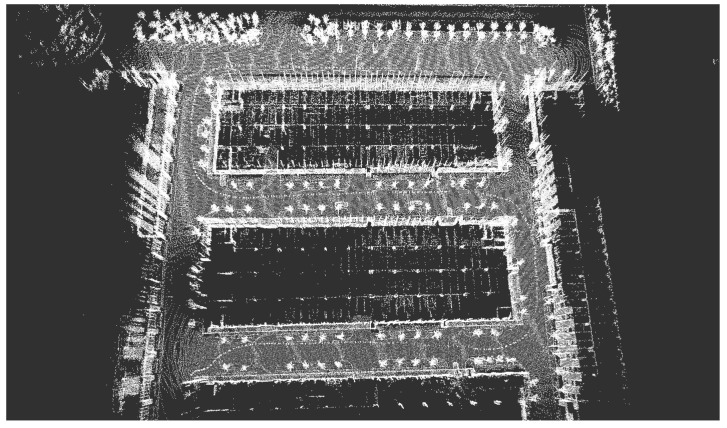
High-precision maps recorded using tracked vehicles equipped with 64-line LiDAR.

**Figure 9 sensors-23-00843-f009:**
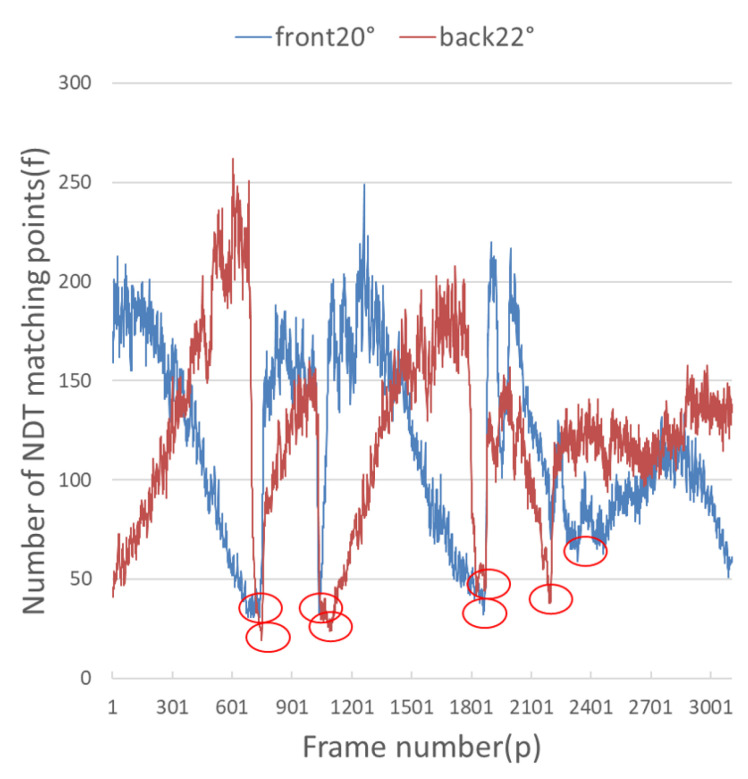
Number of valid points on the front and back during NDT relocation.

**Figure 10 sensors-23-00843-f010:**
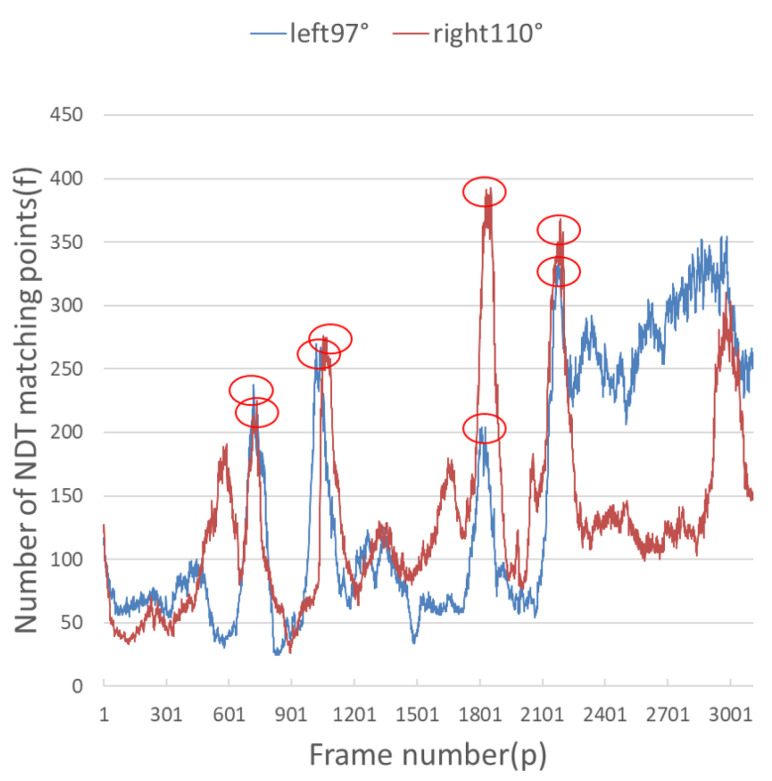
Number of valid points on the left and right during NDT relocation.

**Figure 11 sensors-23-00843-f011:**
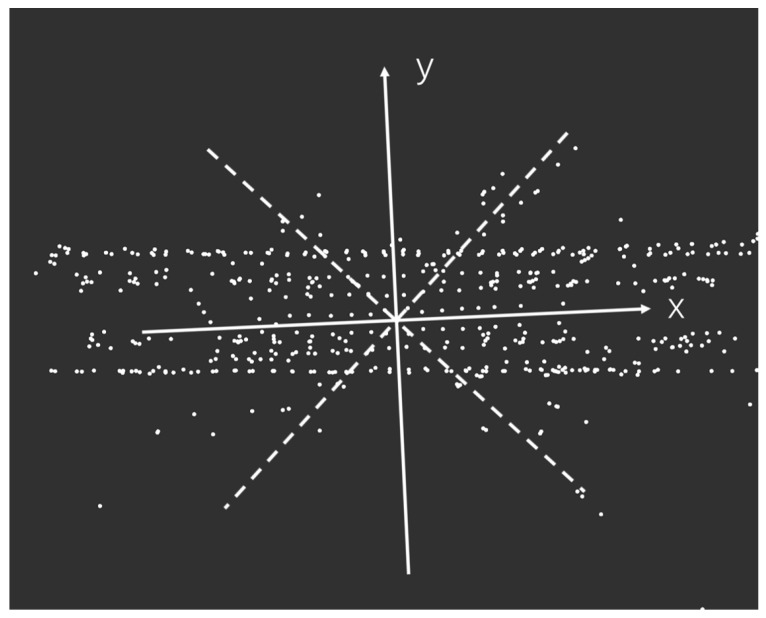
NDT valid matching points on the direct path.

**Figure 12 sensors-23-00843-f012:**
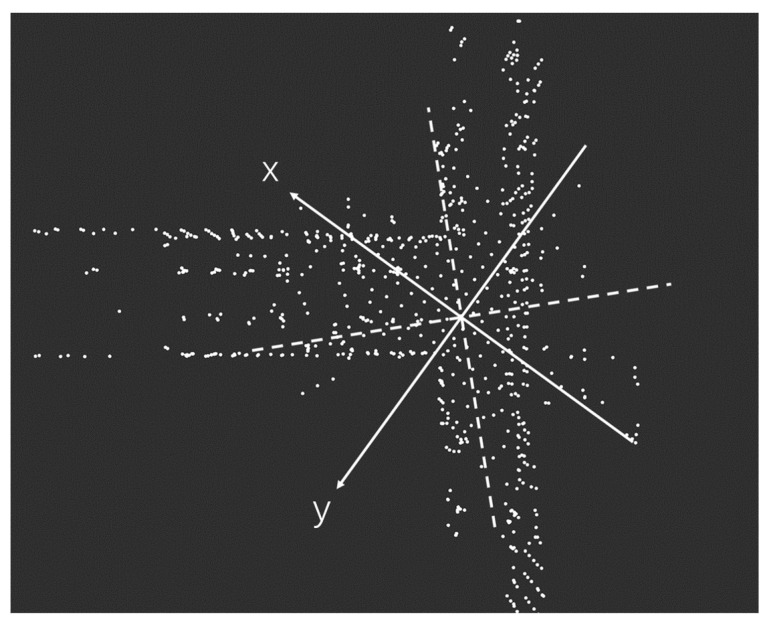
NDT valid matching points on curves.

**Figure 13 sensors-23-00843-f013:**
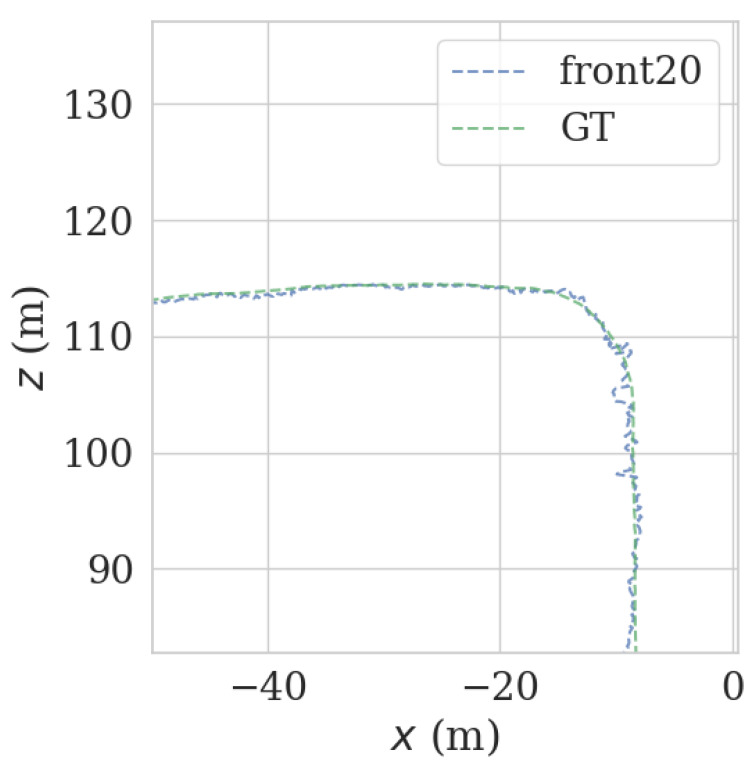
Track error using front LiDAR data at curves.

**Figure 14 sensors-23-00843-f014:**
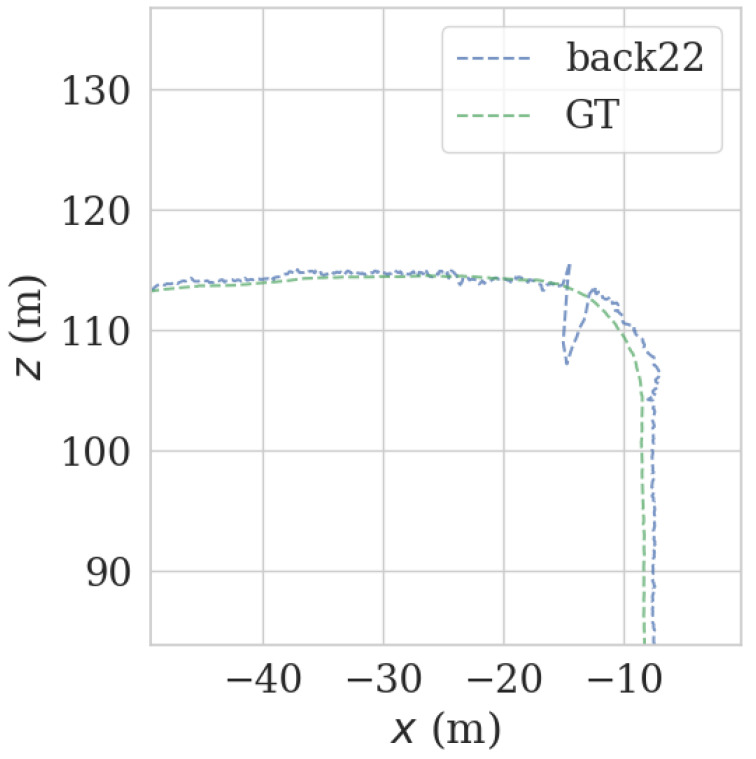
Track error using back LiDAR data at curves.

**Figure 15 sensors-23-00843-f015:**
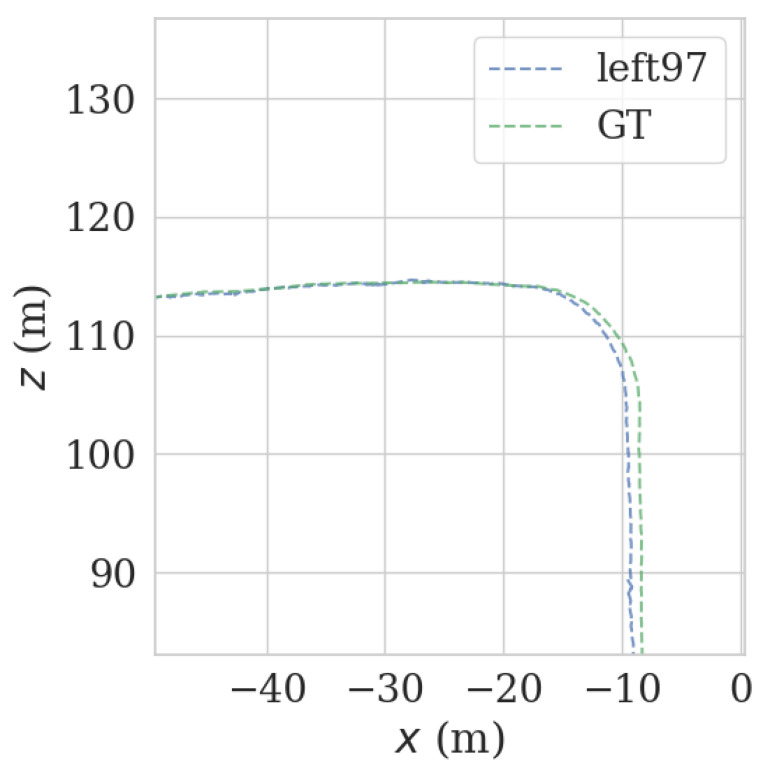
Track error using left LiDAR data at curves.

**Figure 16 sensors-23-00843-f016:**
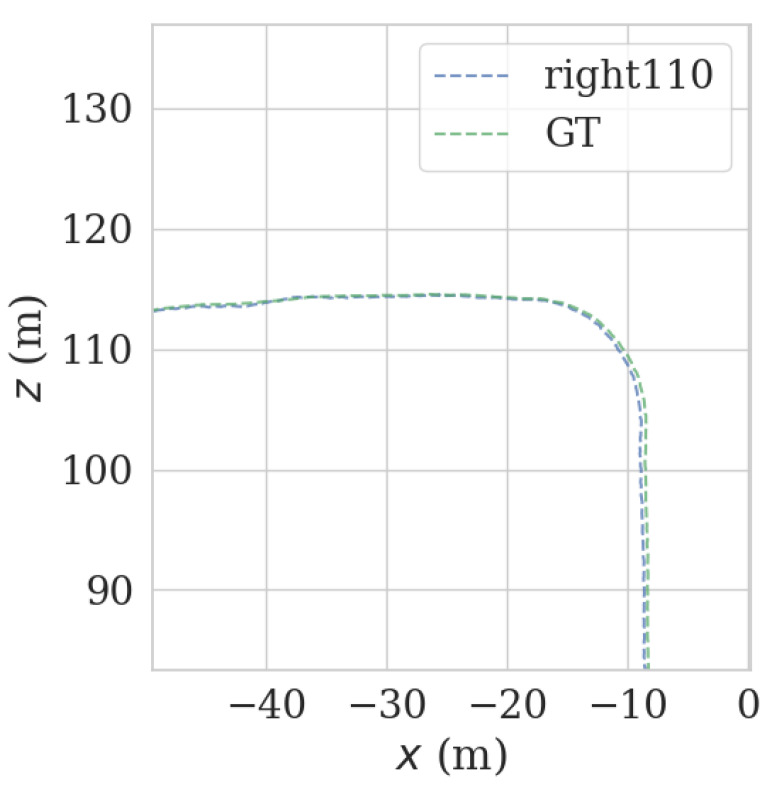
Track error using right LiDAR data at curves.

**Figure 17 sensors-23-00843-f017:**
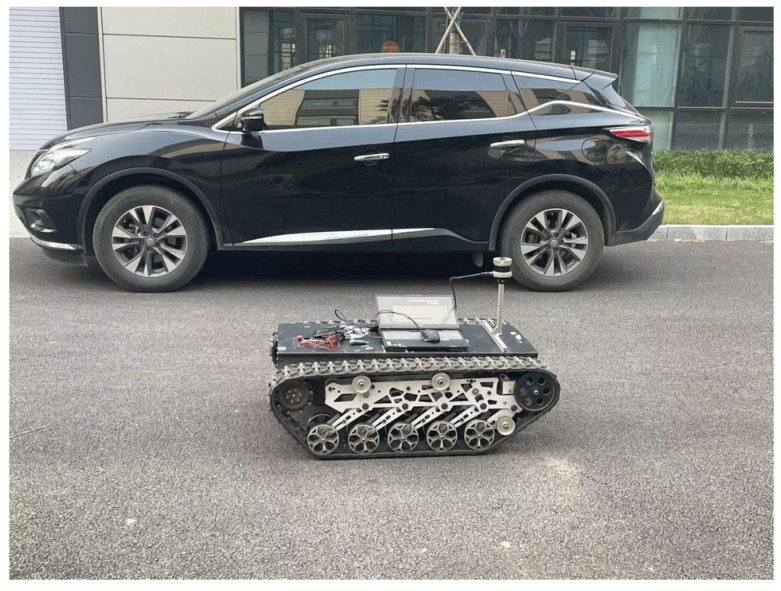
Road conditions with real obstacles on the left side.

**Figure 18 sensors-23-00843-f018:**
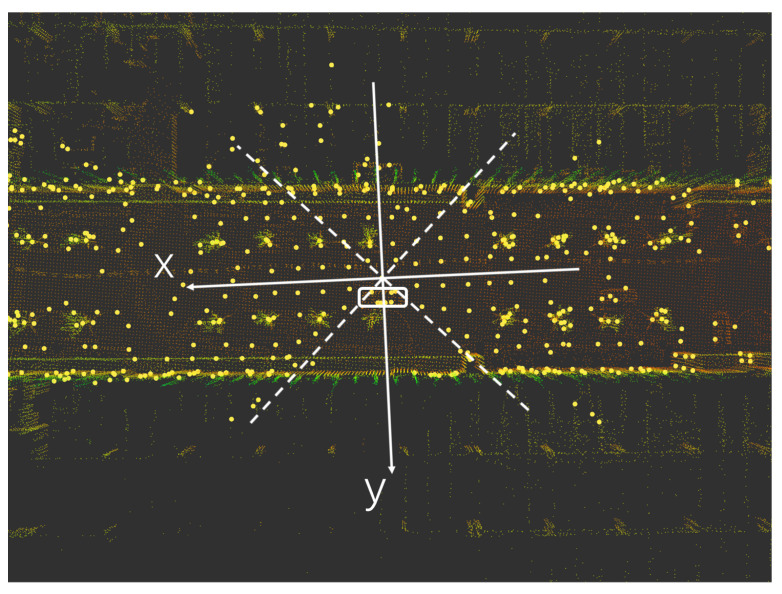
LiDAR point clouds with real obstacles on the left.

**Figure 19 sensors-23-00843-f019:**
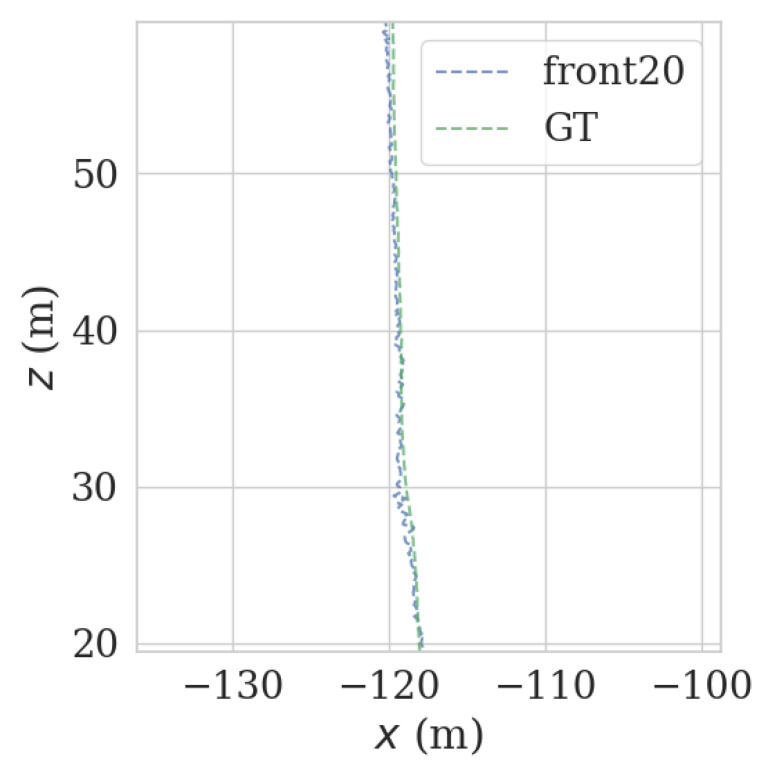
Track error using front-front LiDAR data on the straight track.

**Figure 20 sensors-23-00843-f020:**
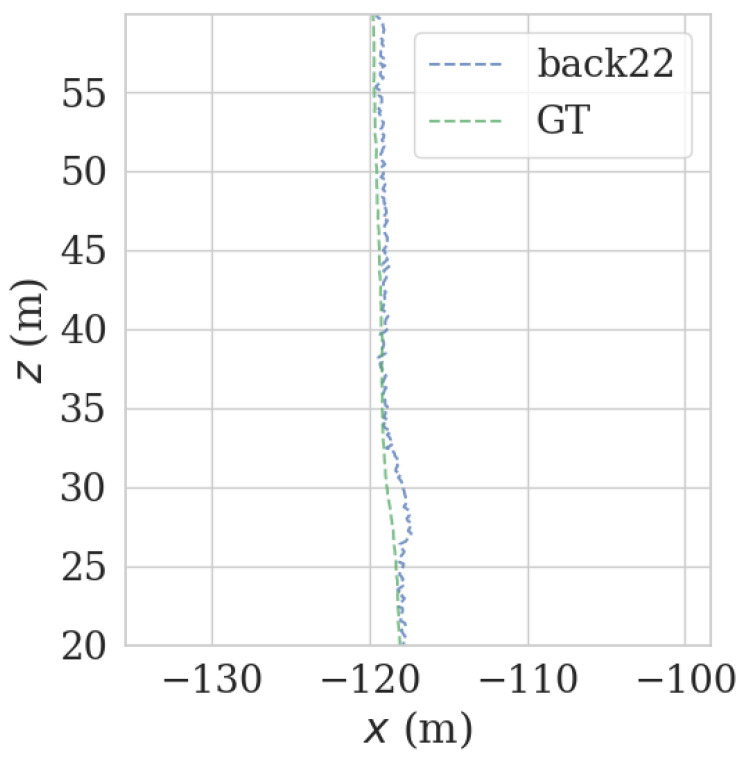
Track error using front-back LiDAR data on the straight track.

**Figure 21 sensors-23-00843-f021:**
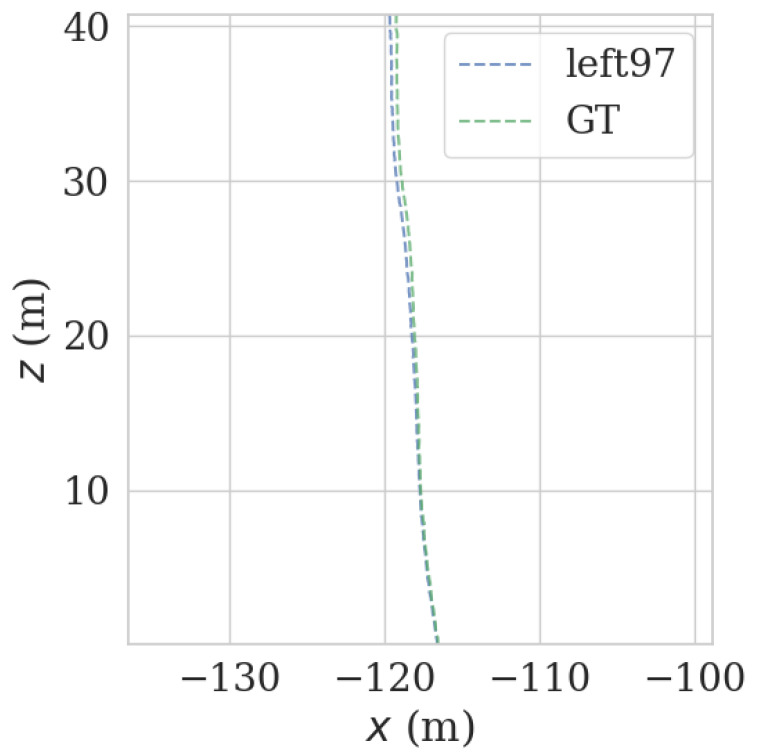
Track error using front-left LiDAR data on the straight track.

**Figure 22 sensors-23-00843-f022:**
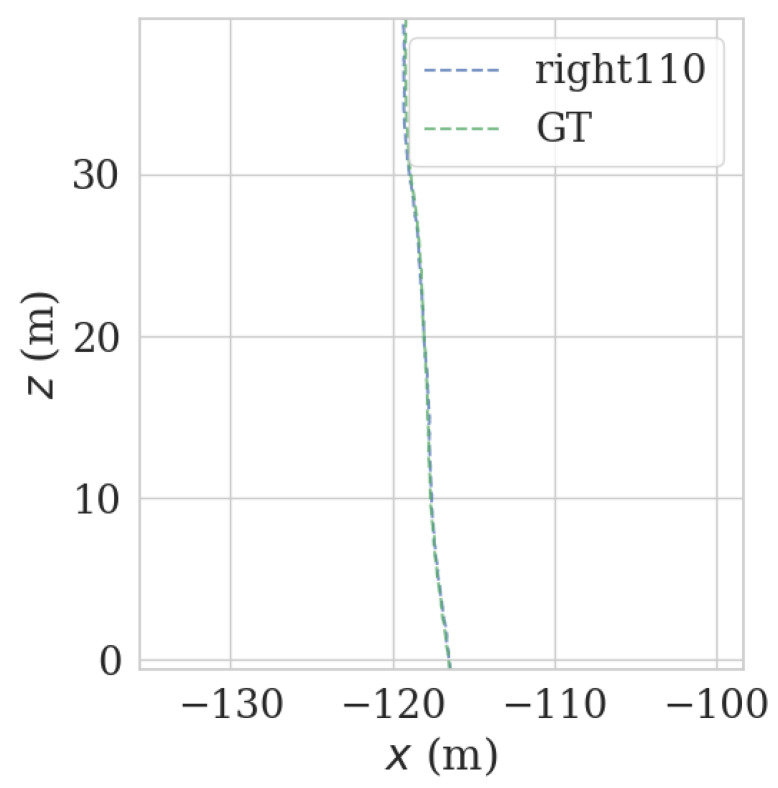
Track error using front-right LiDAR data on the straight track.

**Table 1 sensors-23-00843-t001:** Absolute track error in four directions.

	Max (m)	Mean (m)	Min (m)	Rmse (m)	Num (m)	Ratio (%)
90_front	1.803023	0.820853	0.048090	0.953158	224,995	34.7
180_front	0.468620	0.213139	0.019969	0.224281	316,011	48.7
270_front	0.323754	0.115183	0.010731	0.126000	423,800	65.3
90_back	1.081605	0.646901	0.023327	0.690814	238,414	36.8
180_back	0.930584	0.452172	0.024949	0.500723	332,584	51.3
270_back	0.600951	0.250809	0.023253	0.276960	437,715	67.5
90_left	0	0	0	0	117,606	18.1
180_left	0.492303	0.244453	0.018364	0.265370	337,150	52.0
270_left	0.310869	0.132304	0.014551	0.141764	566,296	87.3
90_right	0	0	0	0	7658	0.12
180_right	1.422529	0.739286	0.022606	0.804200	311,443	48.0
270_right	1.029491	0.498957	0.009679	0.562762	545,708	84.1

**Table 2 sensors-23-00843-t002:** Absolute trajectory error left and right.

	Max (m)	Mean (m)	Min (m)	Rmse (m)	Num (m)	Ratio (%)
90_left	0	0	0	0	117606	18.1
120_left	1.408121	0.533117	0.010324	0.576745	162,363	25.0
150_left	1.042893	0.504791	0.013068	0.549890	223,012	34.5
180_left	0.492303	0.244453	0.018364	0.265370	337,150	52.0
210_left	0.425172	0.164817	0.012010	0.177659	482,429	74.4
240_left	0.374192	0.143122	0.009739	0.154102	534,265	82.4
270_left	0.310869	0.132304	0.014551	0.141764	566,296	87.3
90_right	0	0	0	0	7658	12.2
120_right	0	0	0	0	135,137	20.8
150_right	2.502934	1.440530	0.036937	1.535981	191,471	29.5
180_right	1.422529	0.739286	0.022606	0.804200	311,443	48.0
210_right	1.326894	0.649623	0.014481	0.731533	451,772	69.7
240_right	1.371113	0.661285	0.014154	0.760515	508,636	78.4
270_right	1.029491	0.498957	0.009679	0.562762	545,708	84.1

**Table 3 sensors-23-00843-t003:** Absolute trajectory error of critical values in four directions.

	Max (m)	Mean (m)	Min (m)	Rmse (m)	Num (m)	Ratio (%)
41_front	3.008521	1.307617	0.075811	1.509459	161,495	24.9
33_back	3.445162	1.199947	0.156978	1.278492	154,256	23.8
113_left	5.058614	0.807393	0.025104	0.913465	151,678	23.4
122_right	2.680347	1.295943	0.035288	1.380118	137,933	21.3

**Table 4 sensors-23-00843-t004:** Absolute trajectory error of critical values in four directions.

	Max (m)	Min (m)	Rmse (m)	Num (m)	Ratio (%)
20_front	1.715141	0.021566	0.547751	367,065	14.7
22_back	6.650557	0.018378	1.027503	378,291	15.2
97_left	2.507830	0.031737	0.995748	460,454	18.5
110_right	1.038546	0.023148	0.703556	406,214	16.3

## Data Availability

The data are available upon request.

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
