# Peer review of "A Method of Setting the LiDAR Field of View in NDT Relocation Based on ROI"

_sensors, 2023, doi:10.3390/s23020843_

Round 1
Reviewer 1 Report
Authors showed a novel NDT based LiDAR relocation algorithm and showed some measured data for error rate. I cannot find some broke English grammar mistakes. However, authors need to elaborate some work as mentioned below. Except those comments, the manuscript is well written. After addressing those comments, I can recommend this manuscript to editor.
1. Something wrong between Figures 13-20. at page 12. In addition, labels of x-axis and y-axis are small and unclear.
2. Please use Fig. to Figure.
3. Please use abbreviated journal names in reference section.
4. Please provide city and country information for conference papers.
5. Authors had better show error rate or error distances extracted from the data of Figures 13-20.
6. Authors need to provide the Table which shows comparison data of authors' work and previous other researchers' work before conclusion section.
7. In the Introduction section, description of the previous work is limited. Please introduce more examples of LiDAR detection alogrithm based on NDT.
8. Need to define LOAM before using it. Just like full name of LOAM.
9. Authors had better summarize the measured results in Table.
10. If tracking device does not follow the target, how to control the device to relocate the device. Is there any suggestion?
11. In conclusion section, authors had better mention the future work or direction of the research with limitation of the proposed idea.
Author Response
我们非常感谢您最喜欢的考虑和审稿人对我们题为“基于 ROI 的无损检测重定位中激光雷达视野设置方法”(ID:sensors-2041718)的有见地的评论。这些意见对提高论文的质量和可读性非常有价值,对我们今后的研究也有重要的指导意义。我们仔细研究了评论,并完全根据您的评论修改了论文。我们希望这次修改能够获得通过。与您的意见相对应的主要修订如下:
- 图 13-20 之间有问题。在第12页。此外,x轴和y轴的标签很小且不清楚。
修改后的数字已在第 12 和 13 页被替换。图中的错误已被修改。
- 请用Fig. to Figure。
根据期刊的指导方针,论文中提到的所有数字都写成Figure。如果有必要,我想换成图。
- 请在参考部分使用缩写的期刊名称。
参考文献已根据需要进行了修订。第 14 至 15 页显示了更改。
- 请提供会议论文的城市和国家信息。
第 14 至 15 页显示了更改。
- 作者最好显示从图 13-20 的数据中提取的错误率或错误距离。
表 4 总结了图 13 至 22。
- 作者需要在结论部分之前提供表格,该表格显示了作者的工作与以前其他研究人员的工作的比较数据。
表 4 为结论前的表格。目前很少有研究工作做过类似的实验,所以没有放在这里。
- 在引言部分,对以前工作的描述是有限的。请介绍更多基于无损检测的激光雷达检测算法实例。
增加了对NDT算法的更多描述,并举例说明了NDT算法的实用性。从第 53 行到第 59 行,第 2 页。
- 使用前需要定义LOAM。就像LOAM的全称。
LOAM 在 2.1 节中有更详细的定义。土。从第 73 行到第 76 行,第 2 页。
- 作者最好在表中总结测量结果。
测量结果已汇总在现有表格中。图 13-22 提供了对表 4 中数据轨迹的更详细分析。
- 如果跟踪设备没有跟随目标,如何控制设备重新定位设备。有什么建议吗?
(1) 第一种方式是利用gps进行绝对定位。
(2)第二种方式是利用NDT算法,将激光雷达获取的点云数据与构建的点云地图进行匹配,得到当前位置和姿态。
- 在结论部分,作者最好在提出的想法的限制下提及未来的工作或研究方向。
未来的研究方向已根据需要进行了修订。从第 307 行到第 309 行,第 14 页。

Reviewer 2 Report
The authors should consider how to present their work in a more readable and self-contained manner. For example, the authors should consider addressing the following problems.
What is ROI stand for? What is RMSE? These two terms appear in the abstract, but the manuscript never defines them.
The problem of repositioning should also be clearly defined and illustrated. After all, not all readers understand this problem.
What is the main contribution of this work? Does this study propose a new algorithm? Has this work experimentally demonstrated key concepts in setting LiDAR field of view?
Summarizing quantitative results to demonstrate the effectiveness of the proposed approach should be considered.
The problem of repositioning should also be clearly defined for readers.
Author Response
We quite appreciate your favorite consideration and the reviewers ’insightful comments concerning our manuscript entitled “The Method of Setting LiDAR Field of View in NDT Relocation Based on ROI” (ID:sensors-2041718). Those comments are very valuable and helpful for improving the quality and readability of our paper, as well as the important guiding significance to our future researches. We have studied the comments carefully and have revised the paper exactly according to your comments. We hope this revision can meet with approval. The main revisions corresponding to your comments are as follows:
- What is ROI stand for? What is RMSE? These two terms appear in the abstract, but the manuscript never defines them.
ROI has been defined in 2.3 as required. From line 181 to line 182, page 5.
RMSE has been defined in 3 as required. From line 319 to line 222, page 8.
- The problem of repositioning should also be clearly defined and illustrated. After all, not all readers understand this problem.
Relocation has been defined in 1 Introduction as required. From line 19 to line 20, page 1.
- What is the main contribution of this work? Does this study propose a new algorithm? Has this work experimentally demonstrated key concepts in setting LiDAR field of view?
The main contribution of this work in this paper is to propose a method of LiDAR layout and field of view selection based on high-precision map normal distribution transformation (NDT) relocation, to solve the problem of large NDT relocation error and position loss when the occlusion field of view is too large. From line 4 to line 7, page 1.
This paper presents a new algorithm to simulate the real Lidar placement environment and the Lidar environment blocked by obstacles. From line 7 to line 9, page 1.
From Table 1 to Table 4, the influence of different size of Lidar field on Lidar positioning accuracy has been demonstrated.
- Summarizing quantitative results to demonstrate the effectiveness of the proposed approach should be considered.
Current research results are difficult to summarize quantitative results to prove the validity of the proposed method, which can be used as a future research direction.
